# A Cross-Sectional Multivariable Analysis of the Quality of Hemodialysis Patients’ Life in Lahore City, Pakistan

**DOI:** 10.3390/healthcare13020186

**Published:** 2025-01-18

**Authors:** Ghosia Islam, Gulzar H. Shah, Nadia Saeed, Jeffery A. Jones, Indira Karibayeva

**Affiliations:** 1College of Statistical Sciences, University of the Punjab, Lahore 54590, Pakistan; ghosiaharis@gmail.com (G.I.); nadia.stat@pu.edu.pk (N.S.); 2Jiann-Ping-Hsu College of Public Health, Georgia Southern University, Statesboro, GA 30460, USA; gshah@georgiasouthern.edu (G.H.S.); jajones@georgiasouthern.edu (J.A.J.)

**Keywords:** quality of life (QoL), hemodialysis, chronic kidney disease (CKD), multivariable analysis, proportional odds models (POMs), Lahore City

## Abstract

**Background/Objectives:** Chronic kidney disease (CKD) is a severe health problem with dire consequences for the quality of life of millions of individuals and their families around the globe. This quantitative study analyzes the factors associated with hemodialysis patients’ quality of life (QoL) in Lahore City, Pakistan. **Methods:** Primary data from a sample of 384 patients were collected through regular visits to the hospital. We employed proportional odds models (POMs) and structural equation models to identify factors associated with the QoL. **Results:** The results revealed significant associations between various factors and patients’ quality of life. While gender showed no association with quality of life, younger age, single marital status, higher education, higher family income, and employment status were associated with a better QoL. Clinical variables such as the absence of diabetes and hypertension and specific laboratory parameters were protective against deteriorating QoL. Physical symptoms like muscle soreness, cramps, and shortness of breath significantly impacted QoL. Social and environmental factors adversely affected patient well-being, including family distress and financial issues. Psychological variables such as anxiety, depression, and fear of death also influenced QoL. **Conclusions:** The findings underscore the importance of holistic, patient-centered care approaches in renal failure management, highlighting the need for tailored interventions to address the diverse needs of dialysis patients and enhance their QoL. Further longitudinal research is recommended to validate these findings and guide the development of targeted interventions for improving patient well-being in hemodialysis settings.

## 1. Introduction

Chronic kidney disease (CKD) is a significant worldwide public health concern [1,2]. The incidence of CKD is increasingly imposing a substantial burden on healthcare in low-income countries [1,3]. There are five stages of kidney disease which depend on the Glomerular Filtration Rate (GFR), an indication of the kidney’s ability to filter excess fluid and waste [4]. When the GFR level drops below 15 mL/min/1.73 m^2^, it indicates inadequate kidney function, regarded as renal failure in patients [5]. Kidney diseases are rapidly increasing in Pakistan due to factors such as late diagnosis, kidney stones, uncontrolled hypertension, and diabetes [6,7].

Studies have shown that poor kidney function leads to poor quality of life (QoL) [8]. Poorer QoL was associated with increased distress, maladaptive coping, negative illness perceptions, and lower self-efficacy [8]. The physical and emotional QoL scores in dialysis patients are approximately half of those of the average population [9]. Therefore, QoL among dialysis patients declines substantially over time. Improvements in renal rehabilitation in dialysis patients can improve their QoL. For instance, Rooij et al. (2022) found that patients experienced a clinically relevant decline in mental and physical health-related quality of life before dialysis initiation, which stabilized after that [10]. Studies have also reported that patients experienced improvements in their general and physical QoL after starting dialysis, particularly when coupled with lifestyle interventions such as intradialytic exercise [11,12,13]. In 2022, a study analyzed the results of range of motion exercises, demonstrating their effectiveness in improving dialysis adequacy, as indicated by the urea retention ratio (*p* = 0.027) and Kt/V (*p* = 0.017) [12].

To our knowledge, no national health information system for tracking CKD is available. Furthermore, the scientific literature lacks information on factors associated with CKD in Pakistan and estimates of the burden of kidney disease and related clinical procedures, such as dialysis, in Pakistan [14]. Globally, a large number of persons undergo dialysis treatment [15], estimated at 2.62 million in 2017 [16,17]. In the United States, the estimate was 680,000 in 2014 [18]. An estimated 37 million adults in the United States alone have chronic kidney disease (CKD), with one in three people being at risk of kidney disease [19]. Efforts to enhance lives through education and awareness about the risk factors of this disease are essential [20,21]. There is evidence that the burden of end-stage renal disease is also increasing rapidly around the globe, although data from low-income countries are rarely available. According to one study, conservative kidney management was available in 124 (81%) of 154 countries, highlighting the global availability of this approach [22]. Worldwide, the median number of nephrologists was 9.96 per million population, which varied with income levels [22].

The current study aims to fill critical gaps in research, given the scarcity of research on hemodialysis patients’ QoL in low-income countries such as Pakistan due to the scarcity of secondary data available for such research. Our study fills this gap through primary data on the topic, guided by Wilson and Cleary’s Model of Health-Related Quality of Life (HRQoL) theoretical framework [23]. This theoretical framework posits that the HRQoL is influenced by perceived health status, which is shaped by clinical symptoms and other factors. The other factors comprise intertwined domains such as patients’ biological and physiological characteristics, their ability to function, and the characteristics of their environment. This study investigates the factors influencing the quality of life (QoL) of patients with renal failure. The study analyzes demographic, clinical, physical, psychological, and social and environmental impacts on kidney dialysis patients’ quality of life. The study also examines the characteristics associated with a greater risk of developing kidney disease. The specific research question addressed in this study is as follows: what demographic, clinical, physical, psychological, social, and environmental factors are associated with the QoL of patients with renal failure on hemodialysis in Lahore, Pakistan?

## 2. Materials and Methods

### 2.1. Research Design and Sampling

This study uses an observational, quantitative, cross-sectional research design. The study was conducted on patients diagnosed with renal failure who were receiving dialysis within all hospitals in Lahore City, Pakistan. All the public and private hospitals constituted the sampling frame, and patients were selected from these hospitals. The consecutive sampling technique was used to select an adequate sample size [24,25].

The adequacy of the sample size was determined by using the following formula [26]:$$n=\frac{Z^{2}V^{2}}{e^{2}}$$$$n=\frac{1.96^{2}\;(0.5)(0.5)}{{\left(0.05\right)}^{2}}$$*n* = 384
where *Z* is abscissa of the normal curve and $V^{2}$ is equivalent to the product of two proportions p and q. For our purpose, both proportions are fixed at 50%. A proportion of 50% reflects a higher degree of variability compared to 20% or 80%. This is because proportions of 20% and 80% suggest that a significant majority either lack or possess the attribute in question, respectively. Since a proportion of 0.5 represents the greatest variability within a population, it is frequently used to calculate a more cautious sample size. This approach typically results in a larger sample size than if the actual variability of the population attribute were applied [26]. Conversely, skewed distributions can lead to significant deviations from normality, even in moderately sized samples [27]. In such cases, a larger sample or a full census may be necessary.

The inclusion criteria focused on cases admitted with a diagnosis of kidney disease and included only adults aged 18 and above. The exclusion criteria encompassed the presence of another severe or progressive condition likely to interfere with study outcomes (e.g., advanced cancer, severe heart failure). Consequently, 350 valid samples were identified from the 384 patients considered.

### 2.2. Instrument

The research team constructed a structured questionnaire in collaboration with the clinic researchers and the medical advisors (nephrologists). The questionnaire was pre-tested with 50 patients from the Jinnah Hospital, Lahore. The pre-testing results were used to identify and drop some items weakly representing the domain they were supposed to represent, including clinical variables kt/v, GFR levels, and creatinine levels. Physical factors, such as physical troubles due to dialysis and noticing the smell of sweat, were also excluded after pre-testing. In this study, the instrument demonstrated adequate internal consistency, with a Cronbach’s alpha of 0.85.

### 2.3. Variables

#### 2.3.1. Dependent Variable

The dependent variable, quality of life, was categorized into three response categories: poor, fair, and sound. These categories were assigned ordinal values (poor = 1, fair = 2, good = 3) to represent the increasing levels of quality of life. A proportional odds model was applied to the dataset, treating the patient’s quality of life status as ordinal and accounting for the outcome variable’s ordinal nature.

#### 2.3.2. Independent Variables

The demographic independent variables encompassed gender (male, female), age in years (<20, 21–30, 31–40, 41–50, 51–60, and >60), marital status (single, married), social class (lower, middle, upper), level of education (under matric, matric (completed 10th grade), intermediate (completed 12th grade) and above), employment status (not active, active), family income level in Pakistani Rupees (PKR) (<PKR 20,000, PKR 20,000–50,000, >PKR 50,000), which is equivalent to (<USD 71.8, USD 71.8–USD 179.5, >USD 179.5) in 2024 United States dollars (USD) [28], family history of CKD (no, yes), and the number of comorbidities (none, 1, 2 or more).

The clinical variables comprised diabetes (absent, present), heart disease (absent, present), hypertension (absent, present), blood urea nitrogen (BUN) in mg/dL (<7, 7–20, >20), hemoglobin in g/dL (<11, 11.1–13, >13), calcium in mg/dL (<8, 8–10, >10.1), albumin in g/dL (<3.5, 3.5–4, >4), and potassium in mEq/L (<3.5, 3.5–5, >5).

Physical well-being was evaluated using a 5-item Likert scale (ranging from “not at all” to “extremely”). The categories of the scale included soreness in muscles, cramps, itchiness in the skin, shortness of breath, faintness or dizziness, nausea or upset stomach, dryness in the mouth, lack of appetite, and sleep satisfaction.

Social and environmental well-being was assessed using a 5-item Likert scale (ranging from “not at all” to “extremely”) and encompassed family distress, work affected by illness, financial issues, effect on daily living activities, marital relationship, access to health services, changes in eating habits, staff behavior, and friends and family support.

Psychological well-being was examined using a 5-item Likert scale (ranging from “not at all” to “extremely”) and included feeling isolated, anxiety, depression, life returning to normal, worry about death, ability to concentrate or remember things, fear of spreading the disease, changes in appearance, feeling frustrated, and acting irritably.

### 2.4. Analysis Techniques

Statistical analyses were performed using SPSS 20 and AMOS 18 [29,30]. The chi-square test was employed to examine the association between two nominal variables and one nominal variable with a second ordinal variable, initially introduced by Karl Pearson in 1900. Ordinal logistic regression was conducted for the multivariable analysis of the ordinal dependent variable (a patient’s quality of life was categorized as poor, fair, or good). This analysis was performed utilizing the proportional odds model (POM), as described by McCullagh in 1980. The proportional odds model (POM) is one of the most frequently used models for ordinal logistic regression [31]. It is worth noting that the proportional odds model has also been referred to as the constrained cumulative logit model [32]. The proportional odds model was fitted to the data using the SPSS software [29]. Summary of original contributions presented in the study is available in Appendix A.

## 3. Results

### 3.1. Demographic and Clinical Variables Description

Table 1 presents the demographic and clinical variables of hemodialysis patients. Most participants were male (60%), within the age bracket of 41–50 years (22%), were married (64%), belonged to the lower social class (47%), had an education level below matric (40%), were unemployed (77%), and reported a family income of less than PKR 20,000. Regarding clinical variables, most participants lacked a family history of chronic kidney disease (59%), had one comorbidity (49%), had diabetes (56%), were free from heart disease (69%), and were diagnosed with hypertension (58%). From laboratory findings, it was observed that most patients had a blood urea nitrogen (BUN) level of <40 mg/dL (79%), hemoglobin level > 13 g/dL (50%), a calcium level within the range of 8–10 mg/dL (61%), an albumin level between 3.5 and 4 g/dL (42%), and potassium levels ranging from 3.5 to 5.0 mEq/L (62%).

### 3.2. Ordinal Logistic Regression of Quality of Life

#### 3.2.1. Demographic Variables

After checking the assumption of the proportional odds model (test of parallel lines), all demographic explanatory variables were included in the multivariate proportional odds model (Table 2). Marital status is significantly associated with quality of life; single patients exhibit five times the odds (adjusted odds ratio (AOR) = 4.806, *p*-value < 0.001) of having a better quality of life compared to married patients. Gender has no association with the patient’s quality of life (*p*-value = 0.533). Illiterate patients have lower odds (AOR = 0.638, *p*-value < 0.001) of a better quality of life than those with intermediate-level education or above. Patients with matric level of education also have 51.1% (AOR = 0.489, *p*-value = 0.017) lower odds of having a good quality of life than those with an intermediate-level education or above. Patients with a family income level of less than PKR 20,000 have 81.3% (AOR = 0.187, *p*-value < 0.001) lower odds of having a good quality of life than patients whose family income level is greater than PKR 50,000. Patients within an age interval of less than 20 have three times the odds (AOR = 3.421, *p*-value = 0.01) of having a better quality of life compared to patients aged 60 years and above. Similarly, patients aged 31 to 40 have three times the odds (AOR = 3.407, *p*-value = 0.003) of having a better quality of life compared to patients aged 60 years and above. Patients who are active in their jobs have four times the odds (AOR = 4.233, *p*-value < 0.001) of having a good quality of life compared to patients who are not actively employed.

#### 3.2.2. Clinical Variables

Table 3 illustrates the findings of the ordinal logistic regression model, examining the influence of clinical variables on the quality of life of hemodialysis patients. Patients without diabetes exhibit twice the odds (AOR = 2.190, *p*-value = 0.001) of experiencing a better quality of life compared to those with diabetes. Similarly, patients without hypertension show three times the odds (AOR = 3.320, *p*-value < 0.001) of having a better quality of life compared to patients with hypertension. Furthermore, patients with a hemoglobin level below 11 g/dL demonstrate five times the odds (AOR = 4.943, *p*-value < 0.001) of experiencing a good quality of life compared to those with a hemoglobin level greater than 13 g/dL. Conversely, patients with a hemoglobin level within the range of 11.1–13 g/dL show three times the odds (AOR = 3.040, *p*-value < 0.001) of having a good quality of life compared to patients with a hemoglobin level greater than 13 g/dL. Regarding sodium levels, patients with a level below 135 mEq/L exhibit twice the odds (AOR = 2.153, *p*-value = 0.009) of experiencing a good quality of life compared to those with a level above 148 mEq/L. Similarly, patients with a sodium level within the range of 135–148 mEq/L show two and a half times the odds (AOR = 2.542, *p*-value < 0.001) of having a good quality of life compared to patients with a level above 148 mEq/L. Moreover, patients with an albumin level between 3.5 and 5.0 g/dL demonstrate twice the odds (AOR = 2.112, *p*-value = 0.008) of having a good quality of life compared to those with an albumin level greater than 5 g/dL. Additionally, patients with a blood urea nitrogen level between 7 and 20 mg/dL exhibit three times the odds (AOR = 3.058, *p*-value < 0.001) of experiencing a good quality of life compared to those above 20 mg/dL. Finally, patients with a calcium level between 8 and 10 mg/dL demonstrate twice the odds (AOR = 1.909, *p*-value = 0.028) of having a good quality of life compared to those above 10 mg/dL.

#### 3.2.3. Physical Well-Being

Table 4 presents the results of the ordinal logistic regression model, examining the influence of physical variables on the quality of life among hemodialysis patients. Patients reporting no muscle soreness have four times the odds (AOR = 4.030, *p*-value = 0.041) of experiencing a good quality of life compared to those reporting extreme muscle soreness. Similarly, patients experiencing significant soreness in muscles have three times the odds (AOR = 3.186, *p*-value = 0.008) of having a good quality of life compared to those reporting extreme soreness. Patients without cramps have four times the odds (AOR = 3.754, *p*-value = 0.032) of experiencing a good quality of life compared to those reporting extreme cramping. Additionally, patients with a moderate level of cramps have four times the odds (AOR = 4.495, *p*-value < 0.001) of experiencing a good quality of life compared to those reporting extreme cramping. Likewise, patients without shortness of breath have six times the odds (AOR = 6.265, *p*-value = 0.001) of experiencing a good quality of life compared to those reporting extreme shortness of breath. Patients reporting no faintness or dizziness have five times the odds (AOR = 5.160, *p*-value = 0.006) of having a good quality of life compared to those reporting extreme levels of these symptoms. Patients reporting faintness or dizziness at a somewhat low level have thrice the odds (AOR = 3.397, *p*-value = 0.016) of experiencing a good quality of life compared to those reporting extreme levels. Similarly, patients reporting faintness or dizziness at a moderate level have twice the odds (AOR = 2.718, *p*-value = 0.018) of experiencing a good quality of life compared to those reporting extreme levels. Patients experiencing nausea at a somewhat low level have thrice the odds (AOR = 2.740, *p*-value = 0.026) of having a good quality of life compared to those reporting extreme nausea. Similarly, patients experiencing nausea at a moderate level have twice the odds (AOR = 2.539, *p*-value = 0.037) of having a good quality of life compared to those reporting extreme nausea. Patients reporting no changes in eating habits have three times the odds (AOR = 3.434, *p*-value = 0.011) of experiencing a good quality of life compared to those reporting extreme changes. Additionally, patients reporting changes at a somewhat low level in eating habits have four times the odds (AOR = 4.178, *p*-value = 0.003) of experiencing a good quality of life compared to those reporting extreme changes. Patients reporting changes in eating habits at a moderate level also have twice the odds (AOR = 2.693, *p*-value = 0.037) of experiencing a good quality of life compared to those reporting extreme changes. Regarding sleep satisfaction, patients reporting no or moderate satisfaction have twice the odds (AOR = 2.596, *p*-value = 0.040 and AOR = 2.110, *p*-value = 0.041, respectively) of experiencing a good quality of life compared to those reporting extreme satisfaction. Moreover, patients perceiving the severity of weather conditions at a moderate level have six times the odds (AOR = 6.122, *p*-value < 0.001) of having a good quality of life compared to those perceiving extreme severity. Finally, patients reporting dryness in the mouth at a moderate level have four times the odds (AOR = 4.293, *p*-value = 0.005) of experiencing a good quality of life compared to those reporting extreme dryness.

#### 3.2.4. Social Well-Being

Table 5 presents the findings of the ordinal logistic regression model, investigating the impact of social variables on the quality of life among hemodialysis patients. Patients whose families experienced a lot of distress have two and a half times the odds (AOR = 2.533, *p*-value = 0.004) of reporting a good quality of life compared to those whose families experienced distress at an extreme level. Regarding financial issues, patients with no financial difficulties have three times the odds (AOR = 2.843; *p*-value = 0.017) of reporting a good quality of life compared to those facing extreme financial challenges. Similarly, patients with moderate financial difficulties have three times the odds (AOR = 3.142, *p*-value = 0.003) of reporting a good quality of life compared to those facing extreme financial problems. However, patients with minor financial difficulties have similar odds (AOR = 1.108, *p*-value = 0.010) of reporting good quality of life compared to those with extreme financial problems. In terms of satisfaction with routine activities and marital relationships, moderately satisfied patients have three times the odds of reporting a good quality of life compared to those who are extremely satisfied (AOR = 3.803, *p*-value < 0.001, and AOR = 3.251, *p*-value = 0.002, respectively). Similarly, patients not satisfied with their marital relationship have three times the odds of reporting a good quality of life compared to those who are extremely satisfied (AOR = 3.270, *p*-value = 0.005). Regarding satisfaction with traveling, somewhat satisfied patients have twice the odds of reporting a good quality of life compared to extremely satisfied patients (AOR = 2.449, *p*-value = 0.032). In contrast, moderately satisfied patients have four times the odds (AOR = 3.947, *p*-value < 0.001) of a good quality of life. For friends and family support, patients not satisfied have three times the odds of reporting a good quality of life compared to those who are extremely satisfied (AOR = 3.077, *p*-value = 0.010). Similarly, patients who are somewhat satisfied and moderately satisfied have three times (AOR = 3.340, *p*-value = 0.007) and two times (AOR = 2.437, *p*-value = 0.010) the odds, respectively, compared to those who are extremely satisfied. Finally, regarding satisfaction with staff behavior, patients who are not satisfied have three times the odds of reporting a good quality of life compared to those who are extremely satisfied (AOR = 2.953, *p*-value = 0.001). In comparison, patients moderately satisfied have four times the odds (AOR = 4.441, *p*-value = 0.001) of reporting a good quality of life.

#### 3.2.5. Psychological Well-Being

Table 6 presents the findings of the ordinal logistic regression model, investigating the impact of psychological variables on the quality of life among hemodialysis patients. Patients who do not act irritably towards others have five times the odds (AOR = 5.017, *p*-value < 0.001) of reporting a good quality of life compared to those acting extremely irritably. Similarly, patients who act towards others irritably at a somewhat low level have four times the odds (AOR = 4.091, *p*-value = 0.002). In comparison, those at a moderate level have two times the odds (AOR = 2.330, *p*-value = 0.035) of reporting a good quality of life compared to those acting extremely irritably. Regarding feelings of isolation, patients who do not feel isolated, feel isolation at a somewhat low level, or feel isolation at a moderate level have higher odds (AOR = 4.997, *p*-value = 0.001; AOR = 5.512, *p*-value < 0.001; and AOR = 4.472, *p*-value = 0.001, respectively) of reporting a good quality of life compared to those who feel extreme isolation. Patients with a moderate level of anxiety have twice the odds (AOR = 2.476, *p*-value = 0.033) of having a good quality of life compared to those who feel extremely anxious. Similarly, patients with moderate levels of depression have three times the odds (AOR = 2.547, *p*-value = 0.037) of having a good quality of life compared to those who feel extremely depressed. In terms of hope for normalization of life, patients with a moderate amount of hope have three times the odds (AOR = 3.713, *p*-value = 0.004) of having a good quality of life compared to those with extreme hope. Patients who are not worried at all about death have seven times the odds (AOR = 6.937, *p*-value < 0.001) of having a good quality of life compared to those who extremely fear death. Conversely, patients who are somewhat worried, moderately worried, or very much worried about death have higher odds of having a good quality of life compared to those who extremely fear death (AOR = 2.526, *p*-value = 0.024; AOR = 3.836, *p*-value = 0.002; and AOR = 2.442, *p*-value = 0.030, respectively). Regarding concentration changes, patients who do not report a change or report a somewhat-low-to-moderate level of change have higher odds (AOR = 7.853, *p*-value < 0.001; AOR = 7.014, *p*-value < 0.001; and AOR = 6.889, *p*-value < 0.001, respectively) of having a good quality of life compared to those reporting extreme concentration change. Patients who do not fear spreading the disease have a 64.6% lesser chance (AOR = 0.354, *p*-value = 0.013) of having a good quality of life compared to those who fear it extremely. Finally, patients who are not frustrated or moderately frustrated with kidney disease have higher odds (AOR = 4.128, *p*-value = 0.003 and AOR = 2.903, *p*-value = 0.019, respectively) of having a good quality of life compared to those who feel extremely frustrated.

## 4. Discussion

Chronic kidney disease is a worldwide public health concern due to its grim consequences for individuals and families, and it is also a healthcare burden on healthcare infrastructures [33,34,35,36]. Pakistan is a low-resource country where the management of kidney disease involves government-funded dialysis facilities and charitable organizations that provide subsidized or no-cost dialysis services across the country [37]. A small proportion of the population can afford out-of-pocket expenditures, but most dialysis patients cannot bear the significant financial burdens as the annual cost of hemodialysis in Pakistan substantially exceeds the annual per capita income in the country [38,39]. The present study investigated the factors influencing the quality of life among hemodialysis patients in Lahore, Pakistan. Through an observational, quantitative, cross-sectional research design, we explored demographic, clinical, physical, social, and psychological variables to understand their association with the patient’s quality of life.

The demographic characteristics of the study participants revealed some interesting patterns. Significant predictors of a higher quality of life in this study include being single rather than married, being under 20 years old or between 31 and 40 compared to those aged 60 and above, and having active employment status versus being unemployed. On the other hand, significant predictors of a lower quality of life are having an illiterate or matric level of education compared to an intermediate or higher education and a family income of less than PKR 20,000 compared to an income more than PKR 50,000. Male predominance among hemodialysis patients aligns with global trends in renal failure epidemiology, reflecting the higher prevalence of kidney diseases among men [40]. According to the results of our study, gender does not have a causal relationship with patients’ quality of life. This result contradicts several other studies that have shown that male patients with CKD or on hemodialysis tend to have a lower quality of life [41,42,43]. However, another study that assessed the quality of life among hemodialysis patients in India found that male participants had a higher quality of life than female participants [44]. Further complementing previous studies suggesting older age as a risk factor for poorer quality of life in dialysis patients [42], our findings indicate that younger age groups, particularly those below 40 and below 20, exhibited better quality of life. Marital status emerged as a significant predictor of quality of life, with single individuals exhibiting better outcomes than their married counterparts. Although, according to the literature, socioeconomic factors do not affect the mortality of renal failure patients [45], in our study, correlating with previous research, socioeconomic factors such as education level and family income significantly impacted the quality of life, emphasizing the role of socioeconomic disparities in shaping health outcomes among renal failure patients [46]. Interventions aimed at improving educational attainment and financial stability may mitigate the adverse effects of socioeconomic deprivation on the quality of life in this population [47,48].

Clinical variables, including comorbidities, laboratory parameters, and symptomatology, were evaluated to assess their association with quality of life. Notably, the absence of diabetes and hypertension emerged as protective factors against deteriorating quality of life, highlighting the importance of effective chronic disease management strategies in optimizing patient outcomes. Surprisingly, lower hemoglobin levels of less than 11 g/dL and between 11.1 and 13 g/dL were associated with a higher quality of life, which contradicts the existing literature. Anemia is common among CKD patients and is typically associated with higher direct healthcare costs and a lower quality of life among hemodialysis patients [49]. One possible explanation is that patients with lower hemoglobin levels may receive more intensive clinical care, including targeted treatments for anemia, which may improve their overall perception of well-being despite the physiological implications of anemia. Another potential factor could be cultural differences in how symptoms are reported and perceived in this population. Prior studies have highlighted variations in symptom burden and patient-reported outcomes across different cultural and regional contexts, emphasizing the need for further research to elucidate these discrepancies [48,50]. Furthermore, physiological symptoms such as muscle soreness, cramps, shortness of breath, and nausea exerted a profound impact on quality of life, underscoring the imperative nature of symptom management and palliative care in hemodialysis settings [51]. Interventions targeting symptom relief and improving physical well-being are paramount in regard to enhancing the overall quality of life among dialysis patients [52]. A review of the experience of fatigue in hemodialysis patients shows that fatigue can affect physical and mental well-being and significantly impact quality of life [53]. Fatigue was not included in the physical well-being module of our study. However, future studies should examine the influence of fatigue on quality of life in combination with other factors in hemodialysis patients.

Social and environmental factors play a crucial role in shaping patient experiences and perceptions of quality of life. According to Kao and colleagues, increased social activities were associated with better health-related quality of life [54]. A patient’s moderate satisfaction with their living activities/social activities had a significant effect on their lives and was found to be causally related to the patient’s QoL. Our findings underscore the adverse impact of family distress, financial constraints, and interpersonal relationships on a patient’s well-being, emphasizing the need for holistic psychosocial support services within dialysis units. Previous studies indicate that enhancing social support networks, financial assistance programs, and workplace accommodations can alleviate stressors and promote resilience among hemodialysis patients, thereby enhancing their quality of life [55,56,57,58].

Psychological variables such as anxiety, depression, fear of death, and feelings of isolation emerged as significant predictors of quality of life among dialysis recipients, complementing existing data in the literature on this topic [59,60]. Addressing the psychological distress associated with chronic illness through a holistic combination of medical, psychological, and social interventions is imperative in regard to optimizing patients’ mental health and overall well-being. A systematic review on the effectiveness of psychoeducational interventions on the mental health and quality of life of hemodialysis patients showed their effectiveness not only in the short term but also in the medium term [61]. Moreover, patient-centered interventions targeting coping strategies, resilience-building, and existential support may attenuate existential concerns and enhance patients’ ability to navigate the challenges posed by renal failure [9].

The findings of this study should be interpreted and generalized, keeping in view its limitations. This study uses a cross-sectional design which does not allow for establishing causal relationships. The study also uses data from hospitals in a single city, Lahore City, Pakistan. This study setting might be different from facilities in other cities in smaller towns and cities, limiting this study’s generalizability. However, there are benefits associated with the study design. This study design offers an efficient snapshot of the situation of hemodialysis at a single point in time, which allows for examining associations and generating hypotheses. The design is cost-effective, posing a limited burden on the dialysis patients and facilities providing the care compared to longitudinal designs. Other studies have applied cross-sectional designs to study this topic [62,63].

## 5. Conclusions

In conclusion, our study provides comprehensive insights into the complex determinants of quality of life among hemodialysis patients in Lahore, Pakistan. The results of the study provide evidence that patients with renal failure face increased deficits in regard to demographic, clinical, physical, social, and psychological quality of life. Our findings underscore the importance of holistic, patient-centered care approaches in renal failure management. Tailored interventions addressing the diverse needs of dialysis patients can enhance their quality of life and promote optimal health outcomes in this vulnerable population. Further longitudinal research is warranted to validate these findings and inform the development of targeted interventions aimed at improving patient well-being and quality of life in hemodialysis settings.

## Figures and Tables

**Table 1 healthcare-13-00186-t001:** Descriptive statistics for demographic and clinical variables.

Demographic Variables	N (%)	Clinical Variables	N (%)
Gender		Family history of CKD	
Male	209 (60)	No	205 (59)
Female	141 (40)	Yes	145 (41)
Age		No. of comorbidities	
<20	35 (10)	None	29 (8)
21–30	52 (15)	One	148 (42)
31–40	62(18)	Two or more	173 (49)
41–50	78 (22)		
51–60	73 (20)		
>60	50 (14)		
Marital Status		Diabetes	
Single	125 (36)	Absent	155 (44)
Married	225 (64)	Present	195 (56)
Social class		Heart Disease	
Lower	164 (47)	Absent	243 (69)
Middle	160 (46)	Present	107 (31)
Upper	26 (7)		
Level of Education		Hypertension	
Under matric	116 (33)	Absent	147 (42)
Matric	140 (40)	Present	203 (58)
Intermediate and above	94 (27)		
Employment status		Blood Urea Nitrogen	
Not active	270 (77)	<7 mg/dL	4 (1)
Active	80 (23)	7.1–40 mg/dL	71 (20)
		<40 mg/dL	275 (79)
Family Income (PKR)		Hemoglobin	
<20,000	202 (58)	<11 g/dL	77 (22)
20,000–50,000	115 (33)	11.1–13 g/dL	97 (28)
>50,000	33 (9)	>13.1 g/dL	176 (50)
		Calcium	
		<8 mg/dL	73 (21)
		8–10 mg/dL	214 (61)
		>10.1 mg/dL	63 (18)
		Albumin	
		<3.5 g/dL	115 (33)
		3.5–4 g/dL	148 (42)
		>4 g/dL	87 (25)
		Potassium	
		<3.5 mEq/L	38 (11)
		3.5–5.0 mEq/L	216 (62)
		>5.0 mEq/L	96 (27)

**Table 2 healthcare-13-00186-t002:** Ordinal logistic regression of dialysis patients’ quality of life with demographic variables.

Variables	AOR	*p*-Value
α (1)	0.204	0.002
α (2)	2.288	0.101
Marital status		
Single	4.806	0.000 *
Married	1	-
Gender		
Male	0.865	0.533
Female	1	-
Level of Education		
Illiterate	0.638	0.000 *
Matric	0.489	0.017 **
Intermediate and above	1	-
Family Income (PKR)		
<20,000	0.187	0.000 *
20,000–50,000	0.489	0.1
>50,000	1	-
Age		
<20 years	3.421	0.010 **
21–30 years	1.499	0.347
31–40 years	3.407	0.003 *
41–50 years	1.565	0.235
51–60 years	1.692	0.167
>60 years	1	-
Employment Status		
Active	4.233	0.000 *
Not active	1	-

* = *p*-value < 0.01 ** = *p*-value < 0.05. The goodness of fit statistics: chi-square = 341.178; *p* = 0.07; Nagelkerke R-square = 0.352. AOR: adjusted odds ratio.

**Table 3 healthcare-13-00186-t003:** Ordinal logistic regression of dialysis patients’ quality of life with clinical factors.

Variables	AOR	*p*-Value
α (1)	4.267	0
α (2)	49.898	0
Diabetes		
Absent	2.19	0.001 *
Present	1	-
Hypertension		
Absent	3.32	0.000 *
Present	1	-
Hemoglobin		
<11 g/dL	4.943	0.000 *
11.1–13 g/dL	3.04	0.000 *
>13 g/dL	1	-
Sodium		
<135 mEq/L	2.153	0.009 *
135–148 mEq/L	2.542	0.000 *
>148 mEq/L	1	-
Albumin		
<3.5 g/dL	1.585	0.116
3.5–5.0 g/dL	2.112	0.008 *
>5.0 g/dL	1	-
Blood urea nitrogen		
<7 mg/dL	2.275	0.422
7–20 mg/dL	3.058	0.000 *
>20 mg/dL	1	-
Calcium		
<8 mg/dL	1.63	0.166
8–10 mg/dL	1.909	0.028 **
>10 mg/dL	1	-

* = *p*-value < 0.01, ** = *p*-value < 0.05; The goodness of fit statistics: chi-square = 411.182; *p* = 0.339 Nagelkerke R-square = 0.351; AOR: adjusted odds ratio.

**Table 4 healthcare-13-00186-t004:** Ordinal logistic regression of dialysis patients’ quality of life with physical factors.

Variables	AOR	*p*-Value
α (1)	14.224	0
α (2)	670.484	0
Soreness in muscles		
Not at all	4.03	0.041 **
Somewhat low	1.457	0.431
Moderately	1.864	0.142
Very much	3.186	0.008 *
Extremely	1	-
Cramps		
Not at all	3.754	0.032 **
Somewhat low	2.106	0.113
Moderately	4.495	0.000 *
Very much	1.635	0.253
Extremely	1	-
Shortness of Breath		
Not at all	6.265	0.001 *
Somewhat low	1.952	0.121
Moderately	2.201	0.061
Very much	1.584	0.238
Extremely	1	-
Faintness or dizziness		
Not at all	5.16	0.006 *
Somewhat low	3.397	0.016 **
Moderately	2.718	0.018 **
Very much	1.571	0.217
Extremely	1	-
Nausea or Upset stomach		
Not at all	2.516	0.06
Somewhat low	2.74	0.026 **
Moderately	2.539	0.037 **
Very much	0.922	0.865
Extremely	1	-
Lack of appetite		
Not at all	3.434	0.011 **
Somewhat low	4.178	0.003 *
Moderately	2.693	0.037 **
Very much	1.416	0.399
Extremely	1	-
Sleep		
Not at all	2.596	0.040 **
Somewhat low	1.208	0.648
Moderately	2.11	0.041 **
Very much	0.634	0.214
Extremely	1	-
Weather severity		
Not at all	0.426	0.078
Somewhat low	6.122	0.000 *
Moderately	1.831	0.125
Very much	1.814	0.1
Extremely	1	-
Dryness in mouth		
Not at all	2.216	0.206
Somewhat low	4.293	0.005 *
Moderately	2.112	0.115
Very much	1.098	0.834
Extremely	1	-

* = *p*-value < 0.01 ** = *p*-value < 0.05; The goodness of fit statistics: chi-square = 612.230; *p* = 0.580 Nagelkerke R-square = 0.640; AOR: adjusted odds ratio.

**Table 5 healthcare-13-00186-t005:** Ordinal regression of dialysis patients’ quality of life with social factors.

Variables	AOR	*p*-Value
α (1)	1.462	0.408
α (2)	29.755	0
Family distress		
Not at all	0.661	0.351
Somewhat low	0.882	0.763
Moderately	0.681	0.334
Very much	2.533	0.004 *
Extremely	1	-
Financial difficulties		
Not at all	2.843	0.017 **
Somewhat low	1.108	0.010 **
Moderately	3.142	0.003 *
Very much	1.153	0.626
Extremely	1	-
Performing living activities		
Not at all	1.248	0.672
Somewhat low	1.532	0.244
Moderately	3.803	0.000 *
Very much	0.692	0.287
Extremely	100	-
Marital relationship		
Not at all	3.251	0.002 *
Somewhat low	1.521	0.24
Moderately	3.27	0.005 *
Very much	0.601	0.192
Extremely	1	-
Traveling problem		
Not at all	1.927	0.149
Somewhat low	2.449	0.032 **
Moderately	3.947	0.000 *
Very much	0.913	0.807
Extremely	1	-
Friends and family support		
Not at all	3.077	0.010 **
Somewhat low	3.34	0.007 *
Moderately	2.437	0.010 **
Very much	1.643	0.166
Extremely	1	-
Staff behavior		
Not at all	2.953	0.001 *
Somewhat low	2.009	0.157
Moderately	4.441	0.001 *
Very much	1.154	0.664
Extremely	1	-

* = *p*-value < 0.01 ** = *p*-value < 0.05; The goodness of fit statistics: chi-square = 579.075; *p* = 0.867, Nagelkerke R-square = 0.602; AOR: adjusted odds ratio.

**Table 6 healthcare-13-00186-t006:** Ordinal logistic regression of dialysis patients’ quality of life with psychological factors.

Variables	AOR	*p*-Value
α (1)	30.969	0
α (2)	1061.03	0
Act irritably		
Not at all	5.017	0.000 *
Somewhat low	4.091	0.002 *
Moderately	2.33	0.035 **
Very much	1.223	0.595
Extremely	1	-
Feel isolation		
Not at all	4.997	0.001 *
Somewhat low	5.512	0.000 *
Moderately	4.472	0.001 *
Very much	1.469	0.324
Extremely	1	-
Anxiety		
Not at all	0.867	0.775
Somewhat low	2.009	0.062
Moderately	2.11	0.054
Very much	2.476	0.033 *
Extremely	1	-
Depression		
Not at all	1.087	0.849
Somewhat low	1.541	0.264
Moderately	1.962	0.079
Very much	2.547	0.037 **
Extremely	1	-
Life become normal		
Not at all	1.665	0.32
Somewhat low	3.713	0.004 *
Moderately	1.967	0.121
Very much	1.471	0.374
Extremely	1	-
Death fear		
Not at all	6.937	0.000 *
Somewhat low	2.526	0.024 **
Moderately	3.836	0.002 *
Very much	2.442	0.030 **
Extremely	1	-

* = *p*-value < 0.01; ** = *p*-value < 0.05; The goodness of fit statistics: chi-square = 560.469; *p* = 0.991, Nagelkerke R-square = 0.596; AOR: adjusted odds ratio.

## Data Availability

The original contributions presented in this study are included in the Appendix A. Further inquiries can be directed to the corresponding author.

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
