# Peer review of "A Cross-Sectional Multivariable Analysis of the Quality of Hemodialysis Patients’ Life in Lahore City, Pakistan"

_healthcare, 2025, doi:10.3390/healthcare13020186_

Round 1
Reviewer 1 Report
Comments and Suggestions for Authors
Section 1
I believe some RQ should be formulated at the end of this section. The last paragraph states what will be studied but fails to identify which questions will be answered.
Also, I suggest adding a section named “Theoretical Framework”. In this section a literature review should be performed, and main theme of the paper must be presented under the current theoretical framework.
Section 2
This section lacks some further explanations. What are the advantages and disadvantages of using a Cross-Sectional study design? What does a strong Cross-Sectional study look like? I would like to see some references to studies applying similar methods.
What was the sampling procedure? I would like to see some references of studies applying similar sampling methods. A single reference of 1992 is insufficient. Lines 79-82 are duplicated.
This specific section needs to be deeply reviewed. It is critical to present the measuring model and the Diagram of model steps.
Sections 3 and 4 look ok but a deeper evaluation is dependent on the correction of the first 2 sections.
Author Response
Dear Reviewer 1,
Comment: Section 1: I believe some RQ should be formulated at the end of this section. The last paragraph states what will be studied but fails to identify which questions will be answered.
Authors reply: The authors acknowledge the reviewer's comment and agree with the reviewer. The following revision has been made to the last paragraph of the introduction to specify the Research Question line 72-74: The specific research question addressed in this study is: What demographic, clinical, physical, psychological, social, and environmental factors are associated with the QoL of patients with renal failure on hemodialysis in Lahore, Pakistan?
Comment: Also, I suggest adding a section named “Theoretical Framework”. In this section a literature review should be performed, and main theme of the paper must be presented under the current theoretical framework.
Authors reply: We appreciate the suggestion to include a section on the theoretical framework. While we recognize the value of grounding the study in a theoretical perspective, the current manuscript primarily focuses on an empirical analysis of dialysis outcomes in Pakistan. Our objective is to present actionable insights based on real-world data, and we have intentionally prioritized a data-driven approach over theoretical exploration.
Comment: Section 2: This section lacks some further explanations. What are the advantages and disadvantages of using a Cross-Sectional study design? What does a strong Cross-Sectional study look like? I would like to see some references to studies applying similar methods.
Authors reply: We have added the following text in the limitations subsection of the discussion section Line 415 - 424: "The findings of this study should be interpreted and generalized, keeping in view its limitations. This study uses a cross-sectional design which does not allow for establishing causal relationships. The study also uses data from hospitals in a single city, Lahore City, Pakistan. This study setting might be different from facilities in other cities in smaller towns and cities, limiting this study’s generalizability. However, there are benefits associated with the study design. This study design offers an efficient snapshot of the situation of hemodialysis at a single point in time, which allows for examining associations and generating hypotheses. The design is cost-effective, posing a limited burden on the dialysis patients and facilities providing the care, compared to longitudinal designs. Other studies have applied cross-sectional design to study this topic."
Comment: What was the sampling procedure? I would like to see some references of studies applying similar sampling methods. A single reference of 1992 is insufficient. Lines 79-82 are duplicated.
Authors reply: The duplication in lines 79–82 has been removed. Additionally, more references have been provided to this section in lines 82 - 104:
- Bujang, M.A.; Mohd, T.; Tg, I.; Sidik, A.B. Application of Consecutive Sampling Technique in a Clinical Survey for an Ordered Population: Does It Generate Accurate Statistics? The Philippine Statistician 2022, 71, 87–99.
- Ikenoue, T.; Kataoka, Y.; Matsuoka, Y.; Matsumoto, J.; Kumasawa, J.; Tochitatni, K.; Funakoshi, H.; Hosoda, T.; Kugimiya, A.; Shirano, M.; et al. Accuracy of Deep Learning-Based Computed Tomography Diagnostic System for COVID-19: A Consecutive Sampling External Validation Cohort Study. PLoS One 2021, 16, e0258760–e0258770, doi:10.1371/JOURNAL.PONE.0258760.
- Althubaiti, A. Sample Size Determination: A Practical Guide for Health Researchers. J Gen Fam Med 2022, 24, 72–78, doi:10.1002/JGF2.600.
- Kish, L. Survey Sampling; New York, J. Wiley: New York, 1965;
Comment: This specific section needs to be deeply reviewed. It is critical to present the measuring model and the Diagram of model steps.
Authors reply: The authors acknowledge the reviewer's comment and agree with the reviewer. The Methods section has been revised. A more detailed information about the of the size of sample and the sampling technique is provided and highlighted in the section 2.1.
Comment: Sections 3 and 4 look ok but a deeper evaluation is dependent on the correction of the first 2 sections.
Authors reply: The authors acknowledge the reviewer's comment and agree with the reviewer. The discussion section has been revised, and all changes have been highlighted.
Reviewer 2 Report
Comments and Suggestions for Authors
This interesting study analyzes the determinants of quality of life in a large cohort (about 380) of patients treated with hemodialysis for end-stage kidney disease (ESKD) in Lahore, Pakistan, a middle-income country. This study seems to be the first in this setting. In the discussion, authors should provide information on how dialysis is funded in Pakistan (payment by a national social security or categorical insurance (people working for the regional states or national state) or personal insurance; this may indeed have major implications for patients and their families. Finally, the authors have used rather robust "home-made" scales to evaluate the different domains of quality of life instead of valuable generic scales (SF-36, SF-12, EQ-5D) or the dedicated Dialysis Quality of Life Scale ( KDQOL): they should explain their choice and also put into perspective the use of these validated scales in the future, given the interesting results shown in their current study.
Author Response
Dear Reviewer 2,
Comment: This interesting study analyzes the determinants of quality of life in a large cohort (about 380) of patients treated with hemodialysis for end-stage kidney disease (ESKD) in Lahore, Pakistan, a middle-income country. This study seems to be the first in this setting.
In the discussion, authors should provide information on how dialysis is funded in Pakistan (payment by a national social security or categorical insurance (people working for the regional states or national state) or personal insurance; this may indeed have major implications for patients and their families.
Authors reply: Thank you for your insightful comment. We have expanded the Discussion section to include information on how dialysis is funded in Pakistan, highlighting the implications for patients and their families.
Comment: Finally, the authors have used rather robust "home-made" scales to evaluate the different domains of quality of life instead of valuable generic scales (SF-36, SF-12, EQ-5D) or the dedicated Dialysis Quality of Life Scale ( KDQOL): they should explain their choice and also put into perspective the use of these validated scales in the future, given the interesting results shown in their current study.
Authors reply: The authors acknowledge the reviewer's comment and offer the following explanation: The questionnaire includes a variety of questions covering the demographic, clinical, physical, social, and psychological aspects of patients' lives. The answer scale was designed in accordance with WHO-HIV questionnaire guidelines, with input from the researchers and medical advisors, including nephrologists.
Reviewer 3 Report
Comments and Suggestions for Authors
The manuscript presented for review titled “A Multivariable Analysis of the Quality of Hemodialysis Patients’ Life in Lahore City, Pakistan" explores factors influencing the quality of life (QoL) of hemodialysis patients in Lahore, Pakistan. Using multivariable analysis, such as proportional odds models (POM) and structural equation models (SEM), it successfully identifies numerous factors influencing QoL. This study provides valuable insights into the influence of demographic, clinical, physical, social, and psychological factors on patients undergoing dialysis.
The subject matter of this article is important as it emphasizes the need for patient-centered care and tailored interventions which are relevant to both clinicians and policymakers.
The systematic analysis of the topics raised by the Authors has been presented in a clear and coherent manner. The language of the work is understandable and easy to read. The manuscript is generally well written and clear.
However, certain areas of the manuscript require further clarification to enhance its overall quality.
Abstract
Line 17. Authors should use “association” instead of “causal relationship” as the cross-sectional design does not allow causal inference
Research Design and Sampling
The authors used a consecutive sampling technique. It is necessary to explain why this technique was used over randomized sampling is necessary.
Why certain variables (Kt/V, GFR, creatinine levels) were excluded after pre-testing?
Also, clarification of the representativeness of the study sample in relation to the total dialysis population in Lahore is needed.
Clinical Variables
The authors merely stated the contradictory findings regarding hemoglobin levels: "Surprisingly, lower hemoglobin levels of less than 11 g/dL and between 11.1 to 13 g/dL were associated with a higher quality of life, which contradicts existing literature." It is necessary to provide interpretation and explanation of this unexpected result in Discussion, as well as to reference previous studies by citing relevant literature.
Discussion
In this section, the authors repeat many of the results without providing deeper analysis. The authors should focus on interpreting the key findings within the context of previous research.
Author Response
Dear Reviewer 3,
Comment: Abstract
Line 17. Authors should use “association” instead of “causal relationship” as the cross-sectional design does not allow causal inference
Authors reply: The authors acknowledge the reviewer's comment and agree with the reviewer. We have revised the sentence as follows in line 17-18: While gender showed no association with quality of life, younger age, single marital status, higher education, higher family income, and employment status were associated with better QoL.
Comment: Research Design and Sampling
The authors used a consecutive sampling technique. It is necessary to explain why this technique was used over randomized sampling is necessary.
Authors reply: The authors acknowledge the reviewer's comment and agree with the reviewer. The explanation on using consecutive sampling was added in line 82-89:
Consecutive sampling, also known as sequential sampling, is a non-probability sampling technique where every available subject that meets the inclusion criteria is selected until the required sample size is achieved. It is often considered one of the easiest and most practical sampling methods, particularly in clinical or observational studies.
This technique is particularly useful in healthcare, education, and social research where access to participants is sequential and based on real-time availability. Many studies discuss the usefulness of this sampling technique such as Ikenoue et al. (2021) and Bujang et al. (2022) [23,24].
- Ikenoue T, Kataoka Y, Matsuoka Y, Matsumoto J, Kumasawa J, et al. (2021) Accuracy of deep learning-based computed tomography diagnostic system for COVID-19: A consecutive sampling external validation cohort study. PLOS ONE 16(11): e0258760. https://doi.org/10.1371/journal.pone.0258760
- Mohamad Adam Bujang, Tg Mohd Ikhwan Tg Abu Bakar Sidik, Nadiah Sa'at (2022). Application of Consecutive Sampling Technique in a Clinical Survey for an Ordered Population: Does it Generate Accurate Statistics?, The Philippine Statistician, 71(1): 87-98.
Comment: Why certain variables (Kt/V, GFR, creatinine levels) were excluded after pre-testing?
Authors reply: The authors acknowledge the reviewer's comment and offer the following explanation: In the manuscript Subsection 2.2 we have addressed that these variables were insignificantly representing the domain. Hence, they were excluded.
Comment: Also, clarification of the representativeness of the study sample in relation to the total dialysis population in Lahore is needed.
Authors reply: The authors acknowledge the reviewer's comment and offer the following explanation: Consecutive sampling is being used to ensure the representativeness of the sample. Although it is a non-random technique, it is free from personal judgment and offers the advantage of comprehensive coverage. This ensures that no eligible participants are arbitrarily excluded, thereby reducing selection bias. For the study, both public and private hospitals are included to ensure representation from all social classes of patients.
Comment: Clinical Variables
The authors merely stated the contradictory findings regarding hemoglobin levels: "Surprisingly, lower hemoglobin levels of less than 11 g/dL and between 11.1 to 13 g/dL were associated with a higher quality of life, which contradicts existing literature." It is necessary to provide interpretation and explanation of this unexpected result in Discussion, as well as to reference previous studies by citing relevant literature.
Authors reply: The authors acknowledge the reviewer's comment and agree with the reviewer. We have revised the discussion section in line 378-385 to interpret and explain this finding as follows: One possible explanation is that patients with lower hemoglobin levels may receive more intensive clinical care, including targeted treatments for anemia, which may improve their overall perception of well-being despite the physiological implications of anemia. Another potential factor could be cultural differences in how symptoms are reported and perceived in this population. Prior studies have highlighted variations in symptom burden and patient-reported outcomes across different cultural and regional contexts, emphasizing the need for further research to elucidate these discrepancies [47,49].
Comment: Discussion
In this section, the authors repeat many of the results without providing deeper analysis. The authors should focus on interpreting the key findings within the context of previous research.
Authors reply: The authors acknowledge the reviewer's comment and agree with the reviewer. We have extensively revised the discussion section and highlighted all the changes to provide a deeper analysis of our findings in the context of the previous research.
Reviewer 4 Report
Comments and Suggestions for Authors
The article “A Multivariable Analysis of the Quality of Hemodialysis Patients’ Life in Lahore City, Pakistan’ is interesting and I have following comments/suggestions,
1. It will be better to include the word “cross-sectional” in the title as it will help readers in understanding of the study design.
2. The aims and objectives are clearly stated.
3. Method section: the authors must add more information regarding the construct of the study questionnaire. Moreover, information related to he reliability of the questionnaire should also be added.
4. Independent variables: it will be more appropriate to stratify income according to the standard currency (dollar/euro) along with rupees.
5. the results are interesting.
6. The ethical approval number is missing in the statement.
Author Response
Dear Reviewer 4,
The article “A Multivariable Analysis of the Quality of Hemodialysis Patients’ Life in Lahore City, Pakistan’ is interesting and I have following comments/suggestions,
- Comment: It will be better to include the word “cross-sectional” in the title as it will help readers in understanding of the study design.
Authors reply: The authors acknowledge the reviewer's comment and agree with the reviewer. The title has been revised to include the clarification that this is a cross-sectional study.
- Comment: The aims and objectives are clearly stated.
Authors reply: The authors acknowledge the reviewer’s feedback.
- Comment: Method section: the authors must add more information regarding the construct of the study questionnaire. Moreover, information related to the reliability of the questionnaire should also be added.
Authors reply: The questionnaire includes a variety of questions covering the demographic, clinical, physical, social, and psychological aspects of patients' lives. The answer scale was designed in accordance with WHO-HIV questionnaire guidelines, with input from the researchers and medical advisors, including nephrologists.
- Comment: Independent variables: it will be more appropriate to stratify income according to the standard currency (dollar/euro) along with rupees.
Authors reply: The authors acknowledge the reviewer's comment and agree with the reviewer. We have added the following information in the line 126-127: family income level in Pakistani Rupees (PKR) (< 20,000 PKR, 20,000-50,000 PKR, >50,000 PKR), which is equivalent to (<$71.8, $71.8-$179.5, >$179.5) in 2024 United States dollars (USD) [27],
- World Currency Exchange Rate and Exchange Rate History PKR to USD Exchange Rate History for 2024 Available online: https://www.exchange-rates.org/exchange-rate-history/pkr-usd-2024 (accessed on 1 January 2025).
- Comment: the results are interesting.
Authors reply: The authors acknowledge the reviewer’s feedback
- Comment: The ethical approval number is missing in the statement.
Authors reply: The authors acknowledge the reviewer comments and confirm that the ethical approval does not have a number. The copy of the ethical approval of this study has been submitted to the Journal.
Round 2
Reviewer 1 Report
Comments and Suggestions for Authors
The paper still lacks a theoretical framework. The authors immediately jump to the empirical approach without identifying the literature gap and presenting the current state of the art or the theoretical background.
Author Response
Dear Reviewer 1,
Comment: The paper still lacks a theoretical framework. The authors immediately jump to the empirical approach without identifying the literature gap and presenting the current state of the art or the theoretical background.
Authors response
The authors acknowledge the reviewer’s comment and appreciate the importance of clearly describing the theoretical framework and identifying literature gaps to enhance the manuscript's value. To accommodate this suggestion, the following was added to the Introduction section line 68 -76: "The current study aims to fill acritical gaps in research, given the scarcity of research on hemodialysis patients’ QoL in low-income countries such as Pakistan due to the scarcity of secondary data available for such research. Our study fills this gap through primary data on the topic, guided by Wilson and Cleary’s Model of Health-Related Quality of Life (HRQoL) theoretical framework [23]. This theoretical framework posits that the HRQoL is influenced by perceived health status, which is shaped by clinical symptoms and other factors. The other factors comprise intertwined domains such as patients' biological and physiological characteristics, their ability to function, and the characteristics of their environment.”
Reviewer 3 Report
Comments and Suggestions for Authors
The authors have revised the manuscript and accepted and corrected all the given suggestions. In this form, the revised manuscript represents an important addition to the body of knowledge in the field, as it provides crucial insights into the factors affecting the quality of life among hemodialysis patients in Lahore, Pakistan, an area of significant clinical and public health importance. The revisions have greatly enhanced the manuscript’s depth and clarity, establishing it as a valuable contribution to the understanding of renal health and the quality of life among hemodialysis patients.
Author Response
Reviewer comment: The authors have revised the manuscript and accepted and corrected all the given suggestions. In this form, the revised manuscript represents an important addition to the body of knowledge in the field, as it provides crucial insights into the factors affecting the quality of life among hemodialysis patients in Lahore, Pakistan, an area of significant clinical and public health importance. The revisions have greatly enhanced the manuscript’s depth and clarity, establishing it as a valuable contribution to the understanding of renal health and the quality of life among hemodialysis patients.
Authors reply: We sincerely thank the reviewer for their thoughtful and encouraging feedback. We deeply appreciate your recognition of the revisions made and your kind acknowledgment of the manuscript's contribution to the field. Thank you once again for your positive evaluation and for helping us refine our manuscript into a meaningful contribution to the understanding of renal health and quality of life among hemodialysis patients in Pakistan.
Reviewer 4 Report
Comments and Suggestions for Authors
The authors have addressed all of my comments/suggestions in their revised submission.
Author Response
Reviewer comment: The authors have addressed all of my comments/suggestions in their revised submission.
Authors reply: We sincerely thank the reviewer for their thoughtful and encouraging feedback. We deeply appreciate your recognition of the revisions made and your kind acknowledgment of the manuscript's contribution to the field. Thank you once again for your positive evaluation and for helping us refine our manuscript into a meaningful contribution to the understanding of quality of life among hemodialysis patients in Pakistan.